# What makes a good metric?
# Evaluating automatic metrics for text-to-image consistency

**Candace Ross, Melissa Hall, Adriana Romero Soriano, Adina Williams**
Meta AI (FAIR Labs)
{ccross,melissahall,adrianars,adinawilliams}@meta.com

## Abstract

Language models are increasingly being incorporated as components in larger AI systems for various purposes, from prompt optimization to automatic evaluation. In this work, we analyze the construct validity of four recent, commonly used methods for measuring text-to-image consistency—CLIPScore, TIFA, VPEval, and DSG—which rely on language models and/or VQA models as components. We define construct validity for text-image consistency metrics as a set of desiderata that text-image consistency metrics should have, and find that no tested metric satisfies all of them. We find that metrics lack sufficient sensitivity to language and visual properties. Next, we find that TIFA, VPEval and DSG contribute novel information above and beyond CLIPScore, but also that they correlate highly with each other. We also ablate different aspects of the text-image consistency metrics and find that not all model components are strictly necessary, also a symptom of insufficient sensitivity to visual information. Finally, we show that all three VQA-based metrics likely rely on familiar text shortcuts (such as *yes*-bias in QA) that call their aptitude as quantitative evaluations of model performance into question.

## 1 Introduction

Text-to-image (T2I) models are becoming increasingly prevalent, leading to a surge in high-quality generated images (Nichol et al., 2021; Ramesh et al., 2022; Yu et al., 2023a). T2I models take text prompts like "*the purple dog is laying across a flower bed*" as input, and generate images that, ideally, will not only be aesthetically pleasing, but also consistent with the text. For example, if the image generated contains a dog, but the dog is not purple nor laying in a flower bed, the generation would be incomplete in an important way. Several evaluation frameworks have recently been devised to automatically evaluate this relationship, *i.e.* the *consistency* between the text prompt and the generated image.

One metric for evaluating the text-image consistency is CLIPScore (Hessel et al., 2021), which uses a CLIP model (Radford et al., 2021) to compute a similarity score between the text caption and the image. Because CLIP does struggle with aspects of visiolinguistic reasoning such as compositionality (Thrush et al., 2022; Yuksekgonul et al., 2022; Yu et al., 2023b), other recent automatic metrics take a more fine-grained approach (Hu et al., 2023; Cho et al., 2023a;b). Each text-image consistency metric relies on an external language model (LM) to generate questions given the text prompt. In the simplest case, the LM might generate questions: "*is there a dog?*", "*is the dog purple?*", "*are there flowers?*", etc. Then, these LM-generated questions are passed to computer vision (CV) models, typically visual question answering (VQA) models, which calculate an overall consistency score by averaging the correct answers to the questions given the image.

Because these recent automatic scoring approaches rely on the simplicity of LMs and the interpretability of CV modules like VQA, they are being increasingly adopted, with new variations on previous metrics being proposed at a rapid pace. In this work, we take stock of where we are, and determine which (if any) of the existing metrics are most informative. To do this, we take a step back and assemble a list of very basic desiderata that an ideal

| Evaluation Metric | Human Interpretable | External Models (Section 2) | Sensitive to Text Properties (Section 3.1, 4) | Sensitive to Image Properties (Section 3.2, 4) | Robust to Known Shortcuts (Section 3.4) |
|---|---|---|---|---|---|
| CLIPScore | ✗ | CLIP | ✓ | ∼ | N/A |
| TIFA | ✓ | LM, VQA | ✓ | ∼ | ✗ |
| VPEval | ✓ | LM, VQA, obj. detector, OCR | ✓ | ∼ | ✗ |
| DSG | ✓ | LM, VQA | ✓ | ∼ | ✗ |

LM = language model    VQA = visual question answering model    OCR = optical character recognition

Table 1: Desiderata for strong and informative text-image consistency metrics for text-to-image models. Criteria with mixed signal are marked with "∼".

automatic metric for T2I consistency should be expected to satisfy; see Table 1. Next, given our set of desiderata, we evaluate four existing text-image consistency metrics to see if they satisfy these ideal properties and find that none of the tested text-image consistency metrics actually satisfy all of them.

We additionally explore the relationship *between* existing metrics and find correlations with CLIPScore are low, suggesting the new text-image consistency metrics may genuinely contribute novel information above and beyond CLIPScore. However, we also measure how well metrics that were proposed earlier correlate with those that were proposed later, such as TIFA for VPEval and DSG, and VPEval for DSG, and find that all three correlate with each other to a medium or strong degree.

We also perform some ablations to better understand how much text and image information are leveraged. Our results provide additional evidence that all text-image consistency metrics have serious weaknesses in that they insufficiently rely on visual information, and also questions are raised about their text abilities as well.

Our results suggest there is ample room to further refine and extend our existing suite of automatic text-image consistency metrics. Until we have a firm idea of what it is that we want our metrics to accomplish, it will continue to be challenging to design adequate metrics. Our work has taken some initial steps towards proposing a handful of minimal desiderata, but future work could also incorporate additional desiderata to help guide the design of better automatic text-image consistency metrics that are more robust and can better evaluate the performance of text-to-image generation models.

## 2  Approach

### 2.1  What makes a good text-image consistency metric?

Metric conceptualization and operationalization have long been a core part of the scientific work of evaluation in ML research fields (Graham, 2015; Welty et al., 2019; Jacobs & Wallach, 2021). In NLP and in CV, such work focuses on everything from designing metrics that better measure their underlying constructs (Howcroft et al., 2020; Blodgett et al., 2021; Kiela et al., 2021; Xiao et al., 2023), to understanding metric correlations (Liu et al., 2023; Sun et al., 2023), from taking into account relevant control experiments (Barbu et al., 2019) to devising better evaluations and evaluation metrics that avoid shortcuts (Geirhos et al., 2020) and other features that may make measurement unreliable or hard to interpret (Jia & Liang, 2017; Gururangan et al., 2018; Tu et al., 2020; Blodgett et al., 2021; Raji et al., 2021; Wang et al., 2022; Banerjee et al., 2023; Cummings et al., 2023; Sinha et al., 2023; Zheng et al., 2023).

In this work, we focus on the construct validity of four existing automatic text-to-image (T2I) consistency metrics, all of which were recently proposed. As a first stab, all four metrics could arguably be deemed construct valid, as they have all been demonstrated to

correlate highly with human judgements, and have additional desirable properties, such as human interpretability. However, we argue there are several additional properties strong T2I text-image consistency metrics should have (see Table 1 for our minimal desiderata), and none of our investigated metrics have all of them.

In general, desiderata for metrics fall into three classes: (i) desiderata that are necessary for every evaluation metric, (ii) desiderata that are necessary when proposing new evaluation metrics, and finally, (iii) nice-to-haves. For our necessary criteria, a text-image consistency metric should be sensitive to images (Antol et al., 2015) and sensitive to text, if it is to measure the consistency between the two (see Section 3.1, Section 3.2 and Section 4). It should also actually measure text-image consistency in a way that is not affected by previously identified, unwanted artifacts or shortcuts (see Section 3.4).

For necessities when proposing a new metric, newly proposed metrics should also improve above and beyond reasonable baselines, including random baselines, and outperform existing alternative metrics, such as CLIPScore (see Table 2). Another desiderata for proposing a new metric is showing that it contributes additional important information, understanding, or contextualization that the previous metric(s) lacked—to explore this point, we also measure how strongly the three text-image consistency metrics correlate with each other and with an existing T2I metric, CLIPScore (see Section 3.3).

Of course, the few desiderata we explore here are not intended to be fully exhaustive. They are intended to be a starting point, a minimal set of properties that automatic text-image consistency metrics should have. We discuss other desiderata that we might also want our text-image consistency metrics to satisfy in Section 5 below.

## 2.2 Evaluation Metrics – Text-Image Consistency

We focus on four metrics – CLIPScore, TIFA, VPEval and Davidsonian Scene Graph (DSG).

**CLIPScore.** CLIP (Radford et al., 2021) is a vision-language model that maps images and text to a feature embedding space. CLIPScore (Hessel et al., 2021) approximates the text-image consistency by using the cosine similarity between the features of the image and the text using CLIP.

**TIFA.** TIFA, or Text-to-Image Faithfulness Evaluation (Hu et al., 2023), uses two primary components – an LM to generate questions from the text prompt and a VQA model to answer the questions using the generated images. The score is computed as the percent of correctly answered questions from the VQA model.

**VPEval.** VPEval (Cho et al., 2023b) generates visual programs from the text prompt using an LM. Where TIFA questions are in natural language The visual programs are executed by 8 modules, such as scale evaluation and counting, that use 3 different vision and vision-language modules including an object detector, OCR model and VQA model.

**Davidsonian Scene Graph (DSG).** Davidsonian Scene Graph (DSG) (Cho et al., 2023a) is similar to TIFA, using LM-generated questions and a VQA model. The key difference is that DSG focuses on addressing inconsistent and hallucinated answers. Questions are counted as correct if and only previous dependencies were also correct. For instance, if the VQA model incorrectly predicts there is not a dog while correctly predicts that the dog is red in the second question, the second question about the red dog is marked incorrect.

## 2.3 Text-to-Image (T2I) Models

We use three state-of-the-art T2I models: (i) a diffusion model that leverages CLIP embeddings via a paid API (`DM w/ CLIP latents`), (ii) a latent diffusion model (LDM) for which we evaluate two checkpoints – `LDM v1` (from a paid API) and `LDM v2` (open-sourced) – and (iii) the open-source version of `GLIDE`.

## 3   Experiments

| T2I Source | COCO | | | | Winoground | | | |
|---|---|---|---|---|---|---|---|---|
| | CLIP | TIFA | VPEval | DSG | CLIP | TIFA | VPEval | DSG |
| Random Chance | N/A | 43.1 | 39.1 | 33.4 | N/A | 44.0 | 40.6 | 36.0 |
| Real Images | 30.6 | 83.0 | 77.3 | 79.2 | **21.5** | 66.8 | 60.8 | 63.7 |
| GLIDE | **28.9** | **64.7** | **59.6** | **55.4** | **21.5** | 51.1 | 45.6 | 41.5 |
| DM w/ CLIP | **31.7** | **85.3** | **79.1** | **81.5** | 25.0 | 71.0 | 64.6 | 68.4 |
| LDM v1 | 31.2 | 79.2 | 73.3 | 73.9 | 24.4 | 66.3 | 59.4 | 60.5 |
| LDM v2 | 31.0 | 79.1 | 73.4 | 73.6 | 23.5 | 62.7 | 57.5 | 58.1 |

Table 2: Results for 4 text-image consistency metrics evaluated on four T2I models, as well as a random chance baseline. While our analysis is not focused on relative performance between different T2I models, we do **bold** the highest and lowest performing models for each text-image consistency metric.

Before our deeper analysis, we first report the text-image consistency metric scores for five T2I models in Table 2. We use the datasets MS-COCO (Lin et al., 2014) and Winoground (Thrush et al., 2022) for text prompts. In total, we have 11,525 and 769 generated images per model for COCO and Winoground, respectively.[1]

For evaluating CLIPScore, we use the CLIP ViT-L14 checkpoint provided by OpenCLIP (Ilharco et al., 2021). For TIFA, VPEval and DSG, we generate questions using Llama-v2-Chat 70B checkpoint model (Touvron et al., 2023).For ease of comparison, we use the newer BLIP2-Flan T5 XL (Li et al., 2023) for all VQA questions. For the remaining, non-VQA questions in VPEval, we use the same models from their paper. For simplicity, we refer to TIFA, VPEval and DSG as *VQA-based metrics*.

All metrics exceed a naïve random chance baseline. The text-image consistency metrics rank models the same on COCO and Winoground (`DM w/ CLIP` > `LDM v1` > `LDM v2` > ). This consistency suggests either that all metrics are differently informative and their agreement reflects genuine T2I model ranking, or that all metrics actually contribute the same kind of information and are redundant. Section 3.3 below will adduce evidence for the latter interpretation. We also observe that, for all metrics, the generated images from at least one T2I model actually score higher than the real images. This could be because real images are visually richer, denser scenes with more necessary information to process.

### 3.1   Experiment 1: Relationship with linguistic properties

An ideal text-image consistency metric should draw on information from provided text *and* from the image. First, to determine whether the text-image consistency metrics are highly dependent on the text in particular, we evaluate the correlation between the metrics and several standard linguistic properties of the text prompt. We measure Spearman's rank correlation between the text-image consistency metric and the *readability*, *complexity* and *length* of the prompt. For readability, we use the Flesch–Kincaid grade level calculation (Flesch, 1948). Flesch-Kincaid approximates the difficulty of reading a passage, based in part on word and sentence length, with higher scores corresponding to more difficulty. For complexity, we use Yngve scores (Yngve, 1960), which use constituency parsing. Deeper and wider of parse trees means more complex sentences. Finally, for prompt length, we use NLTK's word tokenizer (Loper & Bird, 2002) to get a word count, excluding stopwords.

Results are shown in Table 3a for COCO, and in Table 5a in the Appendix for Winoground. Overall, we find that all four metrics on all models are correlated to a medium-to-strong degree with all linguistic properties for COCO ($-0.4$ to $-0.94$) and with the length property for Winoground. This suggests that these metrics are sensitive to the linguistic properties of the text prompts. In general, VPEval and DSG significantly correlate with nearly all linguistic properties in COCO for nearly all models, whereas TIFA and CLIPScore show weaker effects.

---

[1]We omit all prompts that refer to people.

VQA-based metrics *negatively* correlate with the linguistic properties, probably because "harder" prompts (longer, more complex, higher grade-level) can solicit lower text-image consistency metric scores.[2] CLIPScore is not particularly sensitive to syntactic complexity, supporting prior work that CLIP operates more as a bag of words (Yuksekgonul et al., 2022); this may also explain why it positively correlates with grade-level and prompt length.

| | $\rho$ – Readability (Grade Level) | | | | $\rho$ – Syntactic Complexity | | | | $\rho$ – Length (# of Words) | | | |
| | CLIPScore | TIFA | VPEval | DSG | CLIPScore | TIFA | VPEval | DSG | CLIPScore | TIFA | VPEval | DSG |
|---|---|---|---|---|---|---|---|---|---|---|---|---|
| Real Images | 0.28* | -0.30* | **-0.54*** | **-0.44*** | 0.23* | -0.36* | **-0.46*** | **-0.44*** | 0.29 | **-0.45*** | **-0.76*** | **-0.74*** |
| GLIDE | -0.04 | -0.31* | **-0.63*** | **-0.44*** | -0.01 | -0.37* | **-0.41*** | **-0.41*** | -0.10 | **-0.66*** | **-0.72*** | **-0.77*** |
| DM w/ CLIP | 0.33* | -0.22 | **-0.66*** | **-0.40*** | 0.15* | -0.39* | **-0.45*** | **-0.48*** | 0.28 | **-0.70*** | **-0.80*** | **-0.94*** |
| LDM v1 | **0.49*** | **-0.42*** | **-0.56*** | -0.30* | 0.10 | -0.34* | **-0.51*** | **-0.48*** | **0.50*** | **-0.74*** | **-0.86*** | **-0.91*** |
| LDM v2 | **0.41*** | -0.35* | -0.38* | -0.38* | 0.04 | -0.36* | **-0.50*** | **-0.46*** | **0.42*** | **-0.66*** | **-0.69*** | **-0.84*** |

(a) Spearman's rank correlation between *linguistic properties* and text-image consistency metrics for COCO. Metrics are highly sensitive to linguistic features of the text. Statistically significant values are marked with ∗. Moderate to strong statistically significant correlations are **in bold**.

| | $\rho$ – Concreteness | | | | $\rho$ – Imageability | | | | $\rho$ – ImageNet-21k Caption Overlap | | | |
| | CLIPScore | TIFA | VPEval | DSG | CLIPScore | TIFA | VPEval | DSG | CLIPScore | TIFA | VPEval | DSG |
|---|---|---|---|---|---|---|---|---|---|---|---|---|
| Real Images | 0.04* | -0.03 | 0.10* | -0.02 | -0.06* | 0.00 | 0.08* | 0.00 | -0.05 | -0.03 | 0.23 | 0.0 |
| GLIDE | 0.03* | -0.05* | 0.00 | -0.04* | 0.12* | 0.05* | 0.10* | 0.06* | -0.12 | -0.08 | -0.05 | -0.07 |
| DM w/ CLIP | 0.08* | -0.02 | 0.09* | -0.01 | -0.01 | 0.02 | 0.09* | 0.03* | 0.06 | -0.06 | 0.24 | 0.08 |
| LDM v1 | 0.02 | -0.03* | 0.09* | -0.03* | 0.02 | 0.02 | 0.10* | 0.02 | -0.04 | -0.04 | 0.23 | 0.11 |
| LDM v2 | 0.02 | -0.04* | 0.06* | -0.05* | 0.04* | 0.01 | 0.09* | 0.01 | 0.04 | -0.17 | -0.01 | 0.02 |

(b) Spearman's rank correlation between *visual properties* and text-image consistency metrics . Metrics are *not* sensitive to visual properties we evaluated. Statistically significant values are marked with ∗.

Table 3: Correlation for COCO between the text-image consistency metrics and *linguistic properties* are generally moderate to strong across models, while the correlation between these metrics and *visual properties* are predominantly weak. These results suggest text-image consistency metrics are more language-related than vision related.

## 3.2 Experiment 2: Relationship with visual properties

Next, we analyzed the metrics' relationship with visual features. These include the ***imageability*** (how easily hearing a word leads to creating a mental image, Paivio et al. 1968; Bird et al. 2001), ***concreteness*** (how easily a word can be experiences by the senses, Paivio et al. 1968) and ***overlap with ImageNet-21k (IN-21k) object classes*** (Ridnik et al., 2021). We use imageability ratings from Gao et al. (2023) and concreteness ratings provided by Brysbaert et al. (2014) and average across the words in the sentence [3]. For IN-21k, we compute the percentage of words in the prompt that are also IN-21k objects classes. For all visual property calculations, we exclude stopwords. See Table 3b for COCO results, and 5b in the Appendix for Winoground results.

Because language provides such a strong prior in vision-language tasks like VQA (Zhang et al., 2016; Goyal et al., 2017; Agrawal et al., 2018; Lin et al., 2024), we wanted to ensure the

---

[2]Prompt difficulty affect one components in the evaluation pipeline, or it may cascade. Perhaps the T2I model has trouble generating images from harder prompts, leading to low scores, and a negative correlation. Alternatively, perhaps the LM struggles to questionize hard prompts and/or the VQA model struggles to answer them. For our purposes, the existence of these correlations is sufficient to motivate our conclusions, although future work could try to isolate which subcomponents suffer from insensitivity to visual and/or textual information of text-image consistency metrics.

[3]For concreteness and imageability, we assign any missing words the lowest imageability/concreteness score in the corpus. We also tested out (i) assigning the missing words a score of 0 and (ii) omitting missing words. We observed no discernible difference in the correlations.

visual modality was being used in these metrics. We found essentially no correlation between the text-image consistency metrics and the visual properties we evaluated, suggesting the metrics may insufficiently leverage visual properties.

### 3.3 Experiment 3: How distinct are these metrics?

When a new evaluation metric is proposed, we should first check to be sure that it pushes the state of the art. In other words, it should probe new information (or probe old information in a better way). We investigate the extent to which newer VQA-based metrics convey similar information to existing metrics (CLIPScore) and to each other, quantifying similarity as Spearman's rank correlation. We use Spearman's, because it is a nonparametric test and doesn't make strong assumptions about the shape of the underlying distribution. We generate images for every prompt in COCO and Winoground across the 5 different T2I models and using the real images. Then, we compute the scores for metrics (CLIPScore, TIFA, VPEval and DSG) and then measure the pairwise Spearman's correlation between the metrics. Results are shown in Figure 1.

**VQA-based metrics are strongly and significantly correlated with each other.** This may be due to either 1) similar approaches to generating questions using an LM or 2) similar approaches to evaluation by using VQA. CLIPScore shows the weakest correlations with other metrics, perhaps because it operates using cosine similarity. This leads us to question – how can we ensure new VQA-based evaluation metrics introduce or leverage new information?

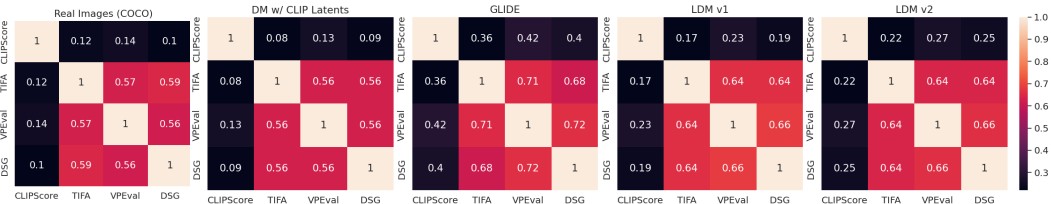

Figure 1: Correlation between each pair of text-image consistency metrics for COCO. The VQA-based metrics are not highly correlated with CLIPScore (excluding GLIDE ), and correlations with real images resemble those from generated images. For VQA-based metrics, correlations are medium to strong and statistically significant, suggesting they may be interchangeable. We observe broadly similar trends for Winoground; see Figure 3.

### 3.4 Experiment 4: Zooming in on the Generated Questions

Given that our results up until this point show text information matters for the metrics, we decided to foreground text-based analyses of the questions for the rest of this section.

**Number of generated questions correlates strongly with text-image consistency metrics.** For this experiment, we analyzed the number of questions generated by the LM. We omit CLIPScore, which uses the caption directly. We find that the number of questions negatively correlates with metric scores, especially for COCO (see Table 4). This behavior makes some sense, because including more questions gives the model more chances to make a mistake.

However, if the correlation between our metrics and basic properties of one of its subcomponents, namely the LM, is strong, we might ask whether having the whole evaluation pipeline genuinely contributes more than just using the LM. In this case, the answer seems fairly clear that the LM may be sufficient, with high significant correlations of above $-0.9$ for all models on COCO.[4] While this result is especially stark for COCO, the same trend holds on Winoground, which is a more challenging but smaller dataset, for DSG, but not as strongly

---

[4]Given this strong correlation, we were curious to learn more about this relationship. Visually and mathematically (with a Pearson's correlation that was insignificant), we confirmed that it is not linear, but more research is necessary to fully characterize it.

|  | COCO | | | Winoground | | |
|---|---|---|---|---|---|---|
|  | TIFA | VPEval | DSG | TIFA | VPEval | DSG |
| Real Images | -0.93* | -0.93* | -0.93* | -0.32 | 0.16 | -0.54 |
| GLIDE | -0.93* | -0.93* | -0.93* | -0.24 | -0.44 | -0.64* |
| DM w/ CLIP | -0.92* | -0.92* | -0.92* | -0.08 | -0.27 | -0.84* |
| LDM v1 | -0.97* | -0.97* | -0.97* | -0.44 | -0.17 | -0.85* |
| LDM v2 | -0.96* | -0.96* | -0.96* | -0.35 | -0.20 | -0.31 |

Table 4: Spearman's correlation $\rho$ between # of generated questions and the text-image consistency metrics are negative and very large (larger than $-0.9$ for every metric on COCO).

for TIFA and VPEval. These results seem to suggest that the LM question generation stage alone may supply enough signal to infer the text-image consistency metric scores. This would imply that the VQA component can be omitted from the evaluation pipeline.

**Distribution of VQA Questions.**   Next, we aim to determine whether the text-image consistency metrics might be relying on shortcuts, and/or falling prey to unwanted statistical artifacts. First, recall that text-image consistency metrics pipeline relies on two models: (i) an LM which takes in the prompt and outputs a set of questions and their "ground truth" answers, and (ii) a VQA model that takes in the LM-generated questions and the image generated by the T2I model, and generates answers. The VQA model's answers are then matched to the LM's "ground truth" answers to get a score. Recall that the LM is prompted, by design, to generate binary questions with "yes" answers, or, for TIFA and VPEval, 4-option multiple choice questions with first-correct answers (see Table 6).

Yes- or first-correct ground truth answers is problematic when the pipeline includes a VQA model. Since Antol et al. (2015) originally proposed the VQA as a task, statistical biases in VQA have been a major topic of research (Agrawal et al., 2018; Ray et al., 2019; Shah et al., 2019; Agarwal et al., 2020; Sheng et al., 2021). It's particularly salient in the field that VQA models suffer from `yes`-bias (Zhang et al., 2016) and `first-answer`-bias, meaning they output these two answers at very high base rates (LMs exhibit similar problems (Benchekroun et al., 2023; Zheng et al., 2023)). This means that the naïve random chance baseline we reported in Table 2 may not actually be at all informative – we need a majority class baseline for the VQA model. Absent that, we genuinely cannot be sure whether a "yes" (or first-correct) output from the text-image consistency metrics pipelines means that the text and image are genuinely consistent, or whether it just means that the VQA model is spuriously generating according to its skewed distribution regardless of the inputs. This fact has two additional implications: (i) a "no" (or non-first answer) will be strictly more informative, and (ii) a program of a few lines that merely prints "yes" (or the first answer) could replace the VQA component entirely, and still yield high text-image consistency metric scores without ever seeing any image or prompt.

A yes- or first-correct only evaluation setup not only benefits from VQA artifacts, it also represents a break from more classical VQA evaluation (Antol et al., 2015; Zhang et al., 2016), where the distribution of ground truth test answers is assumed to match the distribution of the training data (no VQA model, to our knowledge has been purposefully trained to always output "yes", even if they do output it at high rates). One could return to the classical testing set up, whereby the test distribution reflects the underlying – albeit "yes"-skewed, distribution of the VQA system – prompting the LM to generate questions where the ground truth answer should be "no". One could also draw on the classical ML research on class imbalance (He & Ma, 2013; Fernández et al., 2018; Henning et al., 2023), taking answer skew into account when calculating final accuracy, perhaps by resampling answer classes to match the VQA training distribution (Buda et al., 2018).

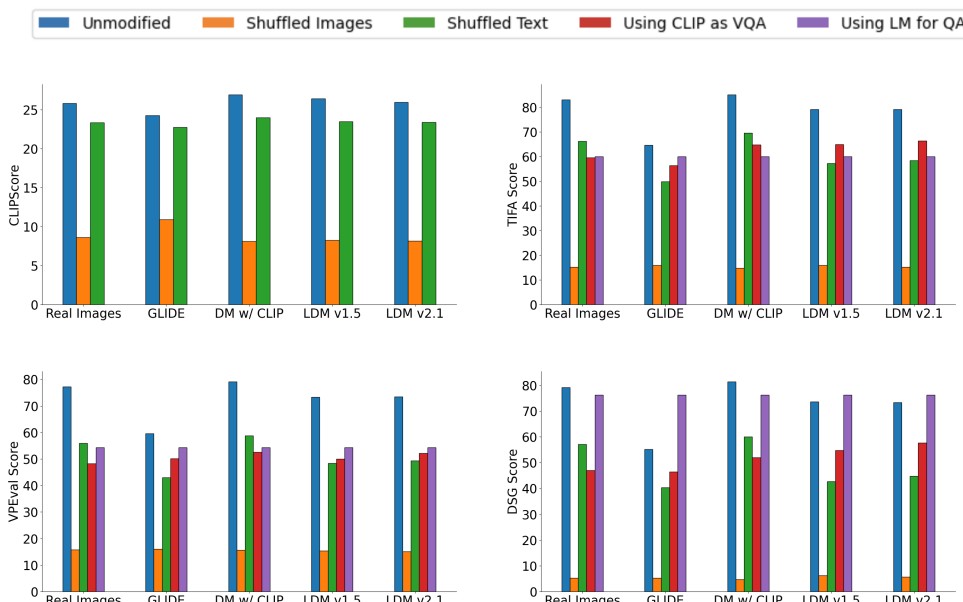

Figure 2: Ablation results for COCO. The bars refer to the original, unmodified metrics **in blue**, shuffled images (`Ablation #1`) **in orange**, shuffled text (`Ablation #2`) **in green**, using CLIP in place of the VQA model (`Ablation #3`) **in red** and using text-only question answering **in purple** (`Ablation #4`). For all metrics, higher is better. While all of the metrics are robust to using random images and to shuffling text, there is a much smaller gap when we ablate the VQA model and instead use CLIP as a proxy (see Figure 5 for an example).

## 4  Ablations: Filling in the Gaps

As we mentioned above, a strong evaluation for T2I models should be sensitive to both the information in the text prompt and the corresponding information in the generated image, but our experiments so far suggest that existing metrics are not actually sufficiently influenced by both text and image. For example, in Sections 3.1 and 3.2, we found that the metrics have moderate to strong correlations with different linguistic properties of the prompt such as readability and prompt length, and much weaker correlations with visual properties like imageability. To explore this further, we perform four ablation analyses to evaluate the degree to which the visual input is leveraged.

Our ablations target the following hypotheses and predictions:

- `Ablation 1:` For each example, we select a random image. If there is no performance degradation, we can conclude that the metrics rely only on the text.
- `Ablation 2:` We reorder the text in the caption (CLIPScore) or questions (TIFA, VPEval, DSG). If there is no performance degradation, we can conclude the metric operates like a bag-of-words.
- `Ablation 3:` We determine whether the VQA model is genuinely necessary by running a pseudo-VQA using CLIP instead. If there is no performance degradation, we can conclude that using a cheaper CLIP alternative is fine.
- `Ablation 4:` Instead of *visual* question answering, we run *text-only* question-answering model using a SOTA QA model. If there is little or no performance degradation, we can conclude that the image is not very important for the metric.

Below, we describe the results of our ablations in turn (also see Figure 2):

**Ablation 1: Shuffled Images.**  We completely ablate the relationship between the images and text by randomly selecting an image from the dataset for each example. Because this shuffling should generally ablate any relationship between questions and images, we expect

text-image consistency metrics to be at or below random chance performance. We observe a huge drop in performance for every metric. This implies that the VQA models generating the text-image consistency metric scores do not completely ignore the visual input. When an image is completely irrelevant, it can throw the VQA model off and solicit a "no", but that doesn't necessarily mean the VQA is sufficiently attending to the image (see Ablation #4 below).

**Ablation 2: Shuffled Text.** Instead of shuffling the text *between* examples, we instead shuffle the text *within* an example (Gauthier & Levy, 2019; Sinha et al., 2021a). The order of the words in each question changes, such that a question "What are the animals in the image?" may become "image animals are what in the the?". Ideally, a strong metric should be sensitive to word order, since it can matter (*e.g.* "is the dog to the left of the cat?", Thrush et al. 2022). All four metrics perform worse on this ablation, meaning they are somewhat sensitive to word order, with CLIPScore being the least sensitive, acting mostly as a bag-of-words.

**Ablation 3: Running VQA using CLIP.** For the VQA-based metrics, we replace the VQA model with CLIP to understand how much VQA itself contributes to the pipeline. For each question with $N$ potential answer choices, we create $N$ captions formatted as "{question}? {answer choice n}". We use CLIP to compute the cosine similarity between the generated image and each of the $N$ captions; see example in Figure 5. We treat the CLIP prediction as correct if the caption with the highest cosine similarity is the caption containing the correct answer. Performance does degrade for the metrics for all T2I models tested, approximately as much as for the shuffled text ablation, suggesting that QA is necessary.

**Ablation 4: Running VQA without the V – Text-only Question Answering.** We ablate the visual input entirely by using a text-only LM for QA. Prior work has shown that VQA models heavily rely on textual priors and can even ignore visual input (Jabri et al., 2016; Goyal et al., 2017; Agrawal et al., 2018). We explore whether these text-image consistency metrics may also ignore the images. To do so, we replace the VQA models with an LM prompted for QA, specifically Flan-T5-XL. We use the same input from the VQA model, formatted as: "Question: {question} Choices: {choices} Answer:". We find that text-only QA performed fairly well, just shy of metrics using VQA. This suggests that VQA may not be strictly necessary: Because the generated questions are likely very skewed towards very probable answers, text-only QA appears to be basically sufficient.

## 5 Discussion and Conclusion

**Other desiderata for automatic text-image consistency metrics.** Ideally, a metric would satisfy all minimal necessary desiderata, and several nice-to-haves as well. There are many additional nice-to-haves that also exist, such as sufficient generation diversity (Hall et al., 2023), robustness to minor input perturbations (Jiang et al., 2020; Gao et al., 2021; Sinha et al., 2021b; Goodarzi et al., 2023), sensitivity to input sample difficulty, etc. Probably most relevant to this work on automatic text-image consistency metrics is compute efficiency. While chaining together submodules – such as LMs, VLMs, or VQA systems – is promising (Mañas et al., 2024), incorporating these subcomponents into model pipelines can add additional compute costs. High compute costs during training have been tied to environmental consequences (Strubell et al., 2019), and it is also possible that incorporating submodules such as LMs into our evaluation pipelines during inference may also have such consequences. Concurrent to our work, Saxon et al. (2024) performed a different type of meta-evaluation and also showed that VQA-based metrics that use additional submodules may not outperform simpler embedding space metrics like CLIPScore. This is why it very important to demonstrate additional utility when proposing novel metrics, especially when they rely on expensive subcomponents.

**Conclusion.** We defined a set of desiderata that should be considered when designing new text-image consistency metrics for text-to-image models. We analyzed 4 metrics – CLIPScore,

TIFA, VPEval and DSG – and found they struggle to meet all desiderata. First, instead of using both the textual and visual information, they rely much more on the text. Next, excluding CLIPScore, they have a very strong correlation with one another, calling into questions how much new information is contributed by each successive metric. Lastly, the VQA-based metrics (TIFA, VPEval and DSG) have very skewed question distributions with artifacts that makes it difficult to know whether they are genuinely measuring text-image consistency. We hope our desiderata can be useful in ensuring we are designing robust evaluation metrics as the field of text-to-image generation continues to grow.

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

# A  Text-image consistency metric Correlations for Winoground

## A.1  Correlations between Linguistic and Visual Properties

We present the results for the correlation between different linguistic and visual properties for text-image consistency metrics in Table 5a and 5b. Similar to the findings for COCO, we observe moderate to strong correlation for some of linguistic properties and essentially no correlation for the visual properties. Unlike COCO, we do not observe correlations for readability and complexity. We hypothesize that this may be because some of the captions in Winoground are less typical (*e.g.* the short captions *truck fire* and *fire truck*).

| | $\rho$ – Readability | | | | $\rho$ – Complexity | | | | $\rho$ – Length | | | |
| | CLIPScore | TIFA | VPEval | DSG | CLIPScore | TIFA | VPEval | DSG | CLIPScore | TIFA | VPEval | DSG |
|---|---|---|---|---|---|---|---|---|---|---|---|---|
| Real Images | 0.06 | -0.05 | 0.10 | -0.10 | 0.06 | -0.10 | 0.02 | -0.18 | 0.06 | -0.26 | -0.05 | **-0.76*** |
| GLIDE | 0.04 | 0.08 | -0.04 | -0.10 | 0.05 | 0.05 | 0.02 | -0.19 | 0.29 | 0.06 | 0.03 | -0.36 |
| DM w/ CLIP | 0.14 | -0.04 | 0.32* | 0.11 | 0.16 | 0.00 | 0.02 | -0.10 | **0.67*** | **-0.38** | **0.44** | **-0.70*** |
| LDM v1 | -0.12 | -0.14 | -0.27 | -0.28 | 0.16 | -0.14 | -0.13 | -0.29* | 0.06 | **-0.51*** | -0.32 | **-0.80*** |
| LDM v2 | 0.04 | 0.37* | 0.16 | 0.37* | -0.08 | -0.20 | -0.20 | -0.23 | **0.45** | 0.08 | **0.50*** | -0.36 |

(a) Spearman's rank correlation between *linguistic features* and text-image consistency metrics on the Winoground dataset.

| | $\rho$ – Concreteness | | | | $\rho$ – Imageability | | | | $\rho$ – IN21k Caption Overlap | | | |
| | CLIP | TIFA | VPEval | DSG | CLIP | TIFA | VPEval | DSG | CLIP | TIFA | VPEval | DSG |
|---|---|---|---|---|---|---|---|---|---|---|---|---|
| Real Images | 0.06 | -0.05 | -0.02 | -0.03 | 0.15* | 0.00 | 0.04 | 0.01 | 0.24 | -0.14 | -0.25 | -0.15 |
| GLIDE | 0.08 | -0.05 | -0.06 | -0.01 | 0.13 | -0.11 | -0.00 | 0.01 | 0.02 | -0.51* | -0.30 | -0.08 |
| DM w/ CLIP | -0.01 | 0.01 | 0.03 | 0.02 | -0.01 | 0.04 | 0.02 | 0.06 | 0.12 | -0.13 | -0.02 | 0.16 |
| LDM v1 | 0.06 | 0.01 | 0.05 | 0.08 | 0.25* | 0.09 | 0.15* | 0.17* | 0.35 | -0.35 | -0.16 | 0.01 |
| LDM v2 | -0.01 | 0.05 | 0.04 | 0.09 | 0.09 | -0.00 | 0.05 | 0.05 | 0.24 | -0.32 | -0.23 | -0.01 |

(b) Spearman's rank correlation between *visual features* and text-image consistency metrics on the Winoground dataset.

Table 5: Spearman's Rank Correlation between text-image consistency metrics and different linguistic properties (top) and visual properties (bottom) for the Winoground dataset. Winoground shows generally simialr trend to COCO, with smaller magnitudes; see Table 3. These findings also support that the metrics are more language-related than vision related.

## A.2  Correlation between Metrics

Next, in Figure 3, we present the correlation between metrics as described in Section 3.3. Similar to COCO, we find moderate to strong correlations between the VQA-based text-image consistency metrics . We also find that these VQA-based metrics do not have a strong correlation with CLIPScore.

# B  Ablation Results on Winoground

We show the results of our four ablations on Winoground in Figure 4. The ablations including shuffling the images between examples (`Ablation #1`), shuffling the text within a given question/caption (`Ablation #2`), using CLIP in place of the VQA model (`Ablation #3`) and lastly using a text-only model for question answering in place of the VQA model (`Ablation #4`). An example of the format for `Ablation #3` where we use CLIP for VQA is shown in Figure 5.

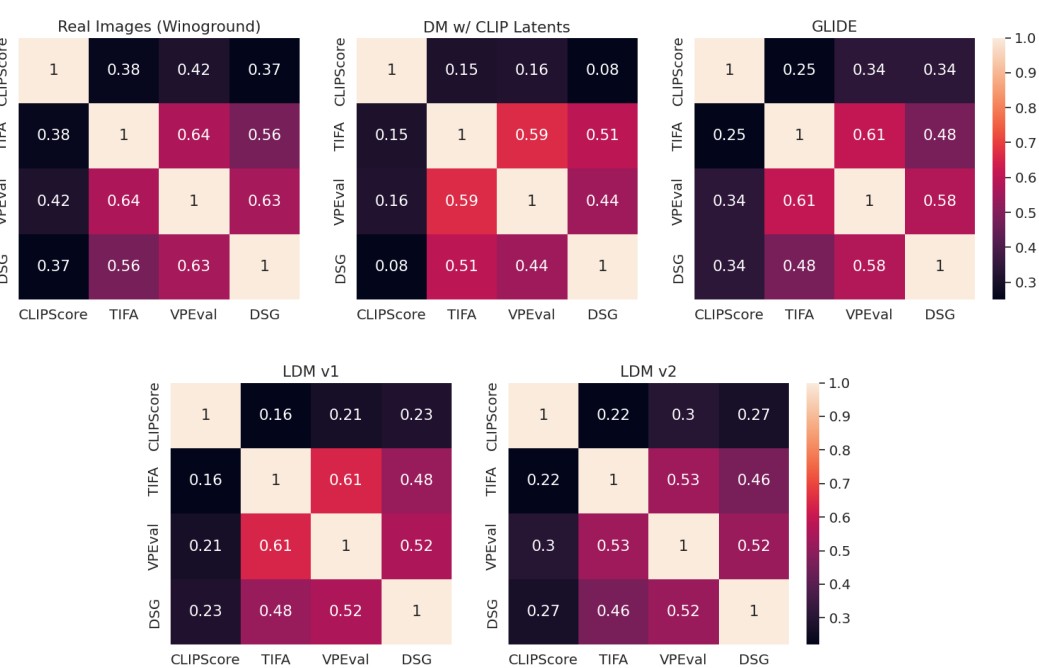

Figure 3: Correlation between each pair of text-image consistency metrics – CLIPScore, TIFA, VPEval and DSG – for 4 text-to-image generative models and for real images. Similar to the finding for COCO shown in Figure 1, we find that VQA-based metrics do not correlate with CLIPScore (again with the exception of GLIDE). For VQA-based metrics on the other hand, correlations are medium to strong and statistically significant. This suggests that the contributions from each metric may not be that distinct from the other metrics. Again similar to COCO, we also observe similar patterns between the real images (top left) and the generated images, suggesting the metrics are likely to be consistent across different image sources such as new text-to-image models. See Section 3.3 for more details.

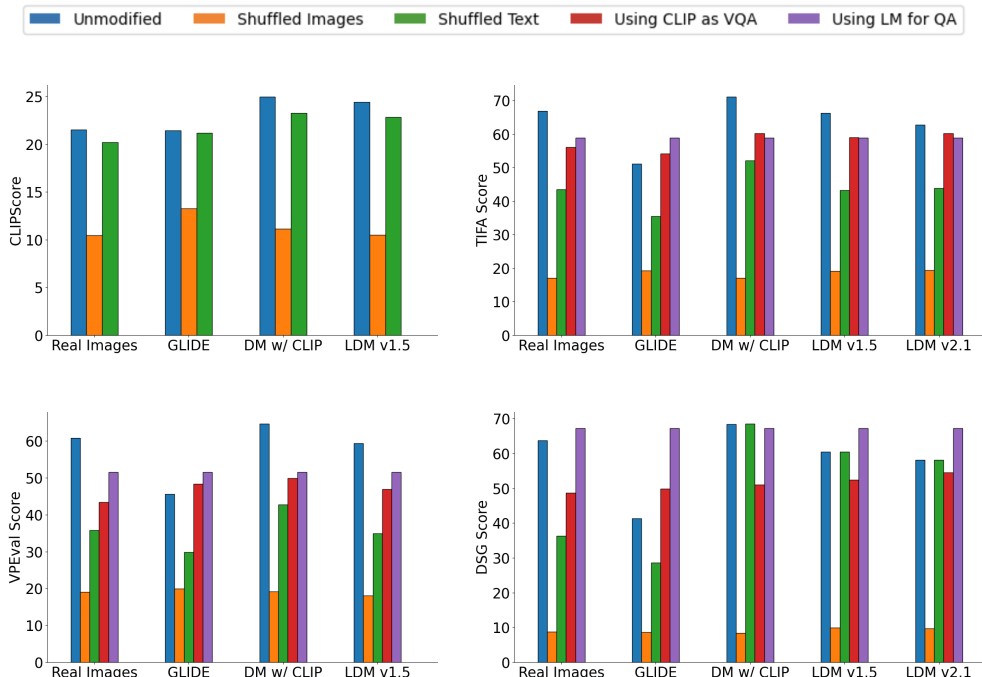

Figure 4: Ablation results for Winoground dataset. The bars refer to the original, unmodified text-image consistency metrics **in blue**, shuffled images (`Ablation #1`) **in orange**, shuffled text (`Ablation #2`) **in green**, using CLIP in place of the VQA model (`Ablation #3`) **in red** and using text-only question answering **in purple** (`Ablation #4`). For all metrics, higher is better.

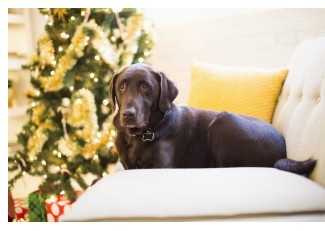

**Question:** What type of animal is this animal?
**Choices:** {dog, cat, bird, fish}
**Answer:** dog

**Captions for CLIP:**
```
c0 = What type of animal is this animal?  dog
c1 = What type of animal is this animal?  cat
c2 = What type of animal is this animal?  bird
c3 = What type of animal is this animal?  fish
```

**Accuracy:** Let $s(c_i)$ be the CLIPScore for a caption $c_i$. If the caption containing the correct choice, $s(c_0)$, is the highest score among the captions, then this question is correct.

Figure 5: For the TIFA, VPEval and DSG text-image consistency metrics, we use ablate the VQA model and replace it with CLIP. Every question is paired with a set of choices and a ground-truth answer **(top)**. The question and choices are combined to form captions, where $N$ choices yields $N$ captions **(middle)**. Finally, the CLIPScore is computed for each of the $N$ captions. If the CLIPScore for the ground-truth caption is the highest among the $N$ captions, then the question is marked as correct **(bottom)**.

## C  Statistics on Question Distribution

In Section 3.4, we dug deeper into the generated questions for TIFA, VPEval and DSG. CLIPScore does not use generated questions, instead using the caption directly, and therefore is not included. Table 6 shows the more detailed statistics of the question distributions. We observe that the distribution is quite skewed with nearly every *yes-no* question having a ground-truth answer of *yes*. Additionally, nearly every multiple-choice question has a first answer bias.

|  | COCO | | | Winoground | | |
|---|---|---|---|---|---|---|
|  | TIFA | VPEval | DSG | TIFA | VPEval | DSG |
| Total # of questions | 61.2k | 118.4k | 63.6k | 1.8k | 7.3k | 1.3k |
| What % are yes/no questions? (%) | 56.8 | 61.5 | 100 | 61.0 | 63.6 | 100 |
| What % is correct answer *yes*? | 99.7 | 99.2 | 100 | 99.3 | 96.6 | 100 |
| What % is correct answer *no*? | <1 | < 1 | 0 | 1.7 | 3.5 | 0 |
| What % are multiple choice? | 43.2 | 38.5 | 0 | 39.0 | 34.4 | 0 |
| What % is correct answer the 1st choice? | 94.0 | 93.8 | N/A | 92.0 | 89.7 | N/A |

Table 6: Statistics on the questions generated by an LM for VQA portion of TIFA, VPEval and DSG. For yes/no questions, the correct answer is *almost always yes* (∼99% of the time). For multiple choice questions (excludes DSG because all questions are binary yes/no), *the first answer is almost always correct*. Overall, the distribution of LM-generated questions for all text-image consistency metrics are highly skewed.

We know the *yes*-bias and *first-correct* bias are a spurious correlations that impacts LMs and/or (V)QA models. Moreover, these spurious lexical correlations from the LM generating the questions and the VQA model answering these questions could compound. Say the LM often writes questions containing the word "bear" and answers them with "yes", regardless of whether the prompt contains "bear" or the generated image contains a bear. Also imagine the VQA model often says "yes" to questions containing "bear". In this case, no matter what the input is, the LM will talk about bears and expect a "yes", and the VQA model will provide it. Surely, this is an extreme toy example, but past work suggests LMs (Shwartz et al., 2020; Tu et al., 2020; Smith & Williams, 2021; Goodarzi et al., 2023) and VQA models (Agrawal et al., 2018; Ray et al., 2019; Shah et al., 2019; Agarwal et al., 2020; Sheng et al., 2021) still suffer from artifacts. If we want to chain models together, we need to think hard about which kinds of spurious correlations may exist.

