# OpenReview forum: "What makes a good metric? Evaluating automatic metrics for text-to-image consistency"
_colmweb.org/COLM/2024/Conference — COLM_

### Official Review · Reviewer_Gt1f · 2024-04-12

**Rating:** 7
**Confidence:** 3
**Ethics Flag:** 1

**Summary:**

Research question of this paper is: "What visio-linguistic properties make a good text-to-image consistency evaluation metric?" The authors identify linguistic metrics (readability, complexity and length) and visual metrics (imageability, concreteness, overlap with large scale image benchmarks), and then meta-evaluate four evaluation metrics:  TIFA, VPEval, DSG and CLIPScore. Finally, the authors analyze the pairwise correlation across metrics and find that three metrics provide new information to CLIPScore.

**Questions To Authors:**

see above

**Reasons To Accept:**

This paper can be a good contribution to COLM. It is well written, well presented and provides an important analysis: How well do we progress in evaluating text-to-image matching? I especially find the identified visio-linguistic properties interesting and necessary (even though incomplete). I especially enjoy reading Ablation: Filling in the gaps section, since it provides insights on how to design better text-to-image evaluation measures (e.g. incorporating information from both modality).

**Reasons To Reject:**

- No major weaknesses are identified. My only minor concern is the abstractness of some visual metrics, such as imageability.

- Do you mind comparing your work against the concurrent preprint: "Who Evaluates the Evaluations? Objectively Scoring
Text-to-Image Prompt Coherence Metrics with (TS2) " (https://arxiv.org/pdf/2404.04251v1.pdf).

- You should settle the real title of your work: Is it:

1) What makes a good metric? Meta-evaluating automatic metrics for text-to-image consistency
2) What makes a good metric? Evaluating automatic metrics for text-to-image consistency

---

> ### Author Rebuttal · Authors · 2024-05-30
>
> Thank you for your comment that this work provides an interesting analysis and that it would be a good contribution to COLM! We address your questions below.
>
> - *abstractness of some visual metrics:* We understand that the abstractness of some of our metrics (e.g. imageability) may seem a little unintuitive. They are however quite standard in the psych/compiling side of things, and since they are underexplored in benchmarks for vision-language modeling, we felt it was important to include them here.
> - *comparison to existing preprint:* Thank you for pointing us to this work! It was a nice read and we’ll absolutely include a discussion of it in our next draft. For example, both their paper and ours address evaluating automatic metrics and consider CLIPScore, but in the end, the two works use different methods but draw  complementary yet different conclusions. We focus on how well metrics correlate with linguistic and visual properties and their failure points via robustness to known shortcuts. They focus on how well metrics can separate faithful and unfaithful image generations using a set of existing image/text prompts. We address inter-metric correlation, but they don’t cover that. Both works do agree that more expensive, VQA-based metrics are not necessarily more robust than simpler ones like CLIPScore. We’ll definitely watch these authors’ contributions carefully as well going forward, as it seems we have similar interests!
>
> Thank you for the note about the two titles, somehow that slipped through editing. In the end, we’re going with “What makes a good metric? Meta-evaluating automatic metrics for text-to-image consistency.” We will update this in OpenReview.

---

> > ### Comment · Reviewer_Gt1f · 2024-06-02
> > **My positivity remains**
> >
> > Dear authors
> >
> > Many thanks for providing an answer, and especially a comparison against the very similar concurrent work. I still believe your paper should be of interest to the community. It works on a relatively under-explored field: evaluating the evaluation metrics. Everyone complains that the evaluation metrics are imperfect but not many attempts to change it. There, I keep my original score.

---

### Official Review · Reviewer_KjKK · 2024-04-29

**Rating:** 5
**Confidence:** 4
**Ethics Flag:** 1

**Summary:**

This paper presents a study to explore the four existing automatic evaluation metrics (CLIPScore, TIFA, VPEval, and DSG) for text-to-image generation. It first designs a set of desiderata, and found none of them meet all the desiderata. And also deliver several findings for the existing metrics: relying much on the text, VQA-based metrics have very skewed question distributions.

**Questions To Authors:**

See my rejection part.

**Reasons To Accept:**

1. It defines a set of classes for desiderata for metrics, which might be useful for the future metrics proposals.
2. It also proposes a couple of data augmentation ways to fill the gaps.
3. And it delivers some interesting findings.

**Reasons To Reject:**

1. My impression to this work is - it empirical explores the four existing metrics in the text-to-image evaluation, and finds some common issues for these metrics, while it does not give a solution yet. It is an interesting paper to explore the existing metrics in the text-to-image evaluation. And the findings in the paper is not surprised. As CLIPScore and other three VQA-based metrics are all pretrained models, they should inherit the bias from their pretraining.  For the second part of this work, it might be better to deliver a new metric or evaluation method based on the proposed desiderata, which may look a more complete work.

Overall, my evaluation to this work is a borderline work, can be either 5 or 6.

---

> ### Author Rebuttal · Authors · 2024-05-30
>
> Thank you for the thoughtful review! We agree that the desiderata are useful for defining metrics and will be useful for future metrics as well.
>
> - *Comment on these results not being surprising:* A primary motivation for our analysis is the fact that these metrics are being widely used right now; if there are issues with them, this affects work on creating SOTA models for T2I generation. These metrics are interestingly even used in much more specific analyses beyond just overall “does this image match this text?” For instance, they’re used in benchmarking the compositional understanding of T2I models [1] and for fairness measurements in generated images [2]. This makes having a comprehensive understanding of these metrics – as well as weaknesses and places for improvement – very important.
> - *Comment on delivering a new metric:* We agree that introducing new metrics are important! Our paper intentionally focuses on the first stage of this process, by defining desiderata and doing a deep-dive into existing metrics to lay the foundation for a follow-up work. Our experiments in Section 4 are designed to robustly identify failure points of existing metrics (for instance, a lack of sensitivity to word order in the text prompt and generated questions) that should be ablated in new metrics. Creating a new metric is an exhaustive process that would warrant a separate paper. For what it’s worth, we are exploring an approach to improve the metrics which relies on generating hard negative questions. We’ve discovered that this approach is promising, but we are still working through some of the specifics. We decided to share issues with current metrics before we have fully solidified our positive metric proposal, because these metrics are really catching on in the field—we want to raise the alarm before the field advances much further on somewhat shaky foundations.
>
> [1] A Contrastive Compositional Benchmark for Text-to-Image Synthesis: A Study with Unified Text-to-Image Fidelity Metrics
>
> [2] DIG In: Evaluating disparities in image generations with indicators for geographic diversity

---

### Official Review · Reviewer_V1Vf · 2024-05-10

**Rating:** 6
**Confidence:** 4
**Ethics Flag:** 1

**Summary:**

This paper evaluates a collection of recent LM + VQA based text-to-image evaluation metrics, and examined whether those approaches are faithfully evaluating some basic desiderata of an ideal text-to-image evaluation metric. Particularly, the author evaluates whether those metrics are human interpretable, and their sensitivity to text properties and image properties, and whether they were robust to known shortcuts. The analysis of those evaluation metric has suggested that those LM + VQA based metric is providing complementary information about the evaluation to the more popular CLIP Score, but they highly correlates between themselves.

**Questions To Authors:**

1. Section 3.4 paragraph 1 "# of generates questions correlates strongly to the text-image consistency metrics" is confusing to me and I don't see how it leads to the conjecture that VQA component can be omitted from the evaluation pipeline. Let's say now we remove the VQA component, and only have the question generation part, how is model going to compute the metric score for text-to-image consistency? Are you suggesting that we directly measuring the "# of generated question" as consistency score? There is no way that this should be the future design of text-to-image eval metric.

2. Most analysis in this paper are statistically, could you also provide some intuition and concrete illustrating examples? For instance, when experiments are designed to show that VQAbased scores are not correlated strongly with CLIPScore, what would be the possible justification over this observation? Could you show examples to justify the reason?

**Reasons To Accept:**

1. The paper is overall clear and the research ideas are straight-forward. The research finding is kind of interesting, suggesting that three existing VQA based evaluations for text-to-image generation are highly correlated, and all having the similar yes-bias.
2. The analysis done for each comparison is convincing, with strong supports from the experiments.
3. The design of experiments are reasonable and solid in most cases, and the ablations are comprehensive.

**Reasons To Reject:**

1. While it has been a pleasant time reading the paper, it does not carry very surprising or significant research discover that reveals any new research direction. It is completely okay to have a small and focused research paper like this but I wouldn't consider it to be the top papers for CoLM.
2. To improve the reading experience, maybe It would be good to have a figure illustrating a generation example and show how it got evaluated by each metrics. It could be even better if that example can also tell the story of the flaws identified with the VQA based approach
3. The proposal of the desiderata future text-to-image metrics should consider is a bit abstract. Especially the "iii) nice-to-haves" is very vague and not well defined.

---

> ### Author Rebuttal · Authors · 2024-05-30
>
> We appreciate your comments that our work is clear, convincing and comprehensive!
> - *overview figure and illustrating examples:* This is a good point about having additional illustrating examples, particularly around the generation. We’ll use the camera ready’s additional page and the appendix. We’ll focus in particular on examples where (i) all metrics perform well (ii) CLIPScore performs well & VQA-based metrics do not (iii) VQA-based metrics do well & CLIPScore does not and (iv) all metrics struggle. We hope this will qualitatively help to interpret each metric’s strengths/weaknesses.
> - *correlation between #/generated questions and metric scores:* Ah, we see how this is a bit confusingly worded. Indeed, you are right, looking only at the #/questions would be a bad T2I metric (according to our desiderata too), and we definitely don't want to propose that. The fact that the #/generated questions appears to affect the measurement, when it shouldn't, suggests the metric is flawed. This is likely because the chances of answering any question in the affirmative is high (as we argue in Section 3.4).
> - *correlation between CLIPScore and VQA-based metrics:* Good question! Here’s our intuition on why VQA-based metrics correlate w/ one another but not with CLIPScore. There are different types of examples where each metric will “thrive”. Let’s take the prompt, “two dogs to the left of a cat”. The cardinality & object position might be difficult for CLIPScore, as CLIP was not trained to understand fine-grained semantics. VQA-based metrics, on the other hand, do leverage fine-grained questions (“how many dogs are there?” “is the cat on the left?”, etc.). Now, let’s consider another prompt, “there’s a white dog and a black dog?” While fine-grained, the generated questions may be underspecified (“Is the dog white?” “Is the dog black?”). The grounding of which dog is being referred to in each question is nontrivial (e.g. the answer to “is the dog white” depends on which dog is being referred to, which is not specified). This is an instance where CLIPScore may perform better. Given these different classes/types of questions where one type of metric struggles where the other may thrive, this could explain the lack of correlation.
> - In terms of your comment on the small, focused nature of our work, we have also addressed a similar concern to reviewer KjKK in “Comment on these results not being surprising”; it might be of interest to you to read our response to them below.

---

### Decision · Program_Chairs · 2024-07-10

**Decision:**

Accept

**Comment:**

This paper examines the construct validity of four automatic text-to-image consistency metrics (CLIPScore, TIFA, VPEval, and DSG). It defines desiderata for such metrics and finds that none of the evaluated metrics satisfy all of them. The paper concludes that these metrics lack sensitivity to language and visual properties, rely on shortcuts, and are highly correlated with each other.

Reviewer Consensus:

Reviewers generally find the paper well-written, clear, and comprehensive. They commend the solid experimental design and the insightful findings, particularly the identification of flaws in existing VQA-based metrics. However, there is a difference in opinion on the significance of the findings and the overall contribution of the work.

Points of Discussion:

- Novelty and Significance: While some reviewers find the research findings interesting and insightful, others suggest the paper lacks a significant research discovery or a novel research direction.
- Desiderata: The proposed desiderata for future text-to-image metrics are considered somewhat abstract and vague, particularly the "nice-to-have" criteria.
- Justification and Examples: Some reviewers request more intuition and concrete examples to illustrate the statistical analyses and findings.
Overall Assessment:

The reviewers agree that the paper tackles an important and relevant topic, and the research is conducted rigorously. However, the lack of a groundbreaking discovery or a clear path for future work leaves some reviewers with reservations.

Recommendation:

Based on the mixed reviews and the lack of strong consensus among the reviewers, my recommendation is to borderline accept this paper.

Suggestions for Improvement:

- Clarity: Refine the proposed desiderata, particularly the "nice-to-have" criteria, to make them more concrete and actionable.
Intuition and Examples: Incorporate more intuitive explanations and illustrative examples to help readers better understand the statistical analyses and results.
- Future Directions: Discuss potential research directions based on the findings, such as the development of new metrics that address the identified shortcomings.
- Comparisons: Include a discussion of the concurrent preprint "Who Evaluates the Evaluations?" to highlight the similarities and differences between the two works.

Incorporating these suggestions would strengthen the paper and make it more valuable to the community.